# A Partial Derivative Approach to the Change of Scale Formula for the Function Space Integral

**DOI:** 10.3390/e23010026

**Published:** 2020-12-26

**Authors:** Young Sik Kim

**Affiliations:** Department of Mathematics, College of Natural Sciences, Industry-University Cooperation Foundation, Hanyang University, 222 Wangsimni-ro, Seongdong-gu, Seoul 04763, Korea; yoskim@hanyang.ac.kr

**Keywords:** fourier transform, directional derivative, change of scale formula, function space, 28C20

## Abstract

We investigate the partial derivative approach to the change of scale formula for the functon space integral and we investigate the vector calculus approach to the directional derivative on the function space and prove relationships among the Wiener integral and the Feynman integral about the directional derivative of a Fourier transform.

## 1. Motivation and Introduction

The solution of the heat (or diffusion)equation:$$-\frac{\partial u}{\partial t}=-\frac{1}{2}\Delta u+V\left(\xi \right)u=Hu\;\;\left(\xi \in R^{d},0\leq t\right),u\left(0,\cdot \right)=\psi \left(\cdot \right)$$
is of the form:(1)$$\begin{array}{ccc} & & u\left(t,\xi \right)=\left(e^{-t\;H}\;\psi \right)\left(\xi \right)=E[\exp\{-(\int_{0}^{t}V\left(x\left(s\right)+\xi \right)ds)\}\psi (x\left(t\right)+\xi )], \end{array}$$
where $\psi \in L_{2}\left(R^{d}\right)$ and $\xi \in R^{d}$ and x(·) is a $R^{d}$—valued continuous function defined on [0,t] such that x(0)=0. *E* denotes the expectation with respect to the Wiener path starting at time t=0 (E is the Wiener integral). H=−Δ+V is the energy operator (or, Hamiltonian) and Δ is a Laplacian and $V:R^{d}\to R$ is a potential. (1) is called the Feynman–Kac formula. Applications of the Feynman–Kac formula (in various settings) have been given in the area of diffusion equations, the spectral theory of the Schrödinger operator, quantum mechanics, statistical physics. (For more details about the application, see [1]).

In [2,3,4,5,6,7,8], formulas for linear transformations of Wiener integrals have been given and the behavior of measure and measurability and the change of scale were investigated and a change of scale formula and a scale invariant measurability were proven.

In [9,10,11], the author proved the change of scale formula on the abstract Wiener space and on the Wiener space and established those relationships in [12] and proved relationships among Fourier Feynman transforms and Wiener integrals for the Fourier transform on the abstract Wiener space in [13]. In [14], the author investigated the partial derivative approach to the integral transform for the function space in some Banach algebra on the Wiener space.

In this paper, we investigate the partial derivative approach and the vector calculus approach to the change of scale formula for the Wiener integral of a Fourier transform and prove relationships among the Wiener integral and the Feynman integral.

## 2. Definitions and Notations

Let $C_{0}\left[0,T\right]$ be the one parameter Wiener space. That is the class of real-valued continuous functions *x* on [0,T] with x(0)=0. Let *M* denote the class of all Wiener measurable subsets of $C_{0}\left[0,T\right]$ and let *m* denote the Wiener measure. $\left(C_{0}\left[0,T\right],M,m\right)$ is a complete measure space and we denote the Wiener integral of a functional *F* by $E_{x}\left[F\left(x\right)\right]=\int_{C_{0}\left[0,T\right]}F\left(x\right)dm\left(x\right)$.

A subset *E* of $C_{0}\left[0,T\right]$ is said to be a scale-invariant measurable provided ρE∈M for all ρ>0, and scale invariant measurable set *N* is said to be scale-invariant null provided m(ρN)=0 for each ρ>0. A property that holds except on a scale-invariant null set is said to hold scale-invariant almost everywhere (s-a.e.). If two functionals *F* and *G* are equal s-a.e., we write F≈G. For more details about the scale-invarant measurability on the Wiener space, see [15].

**Definition** **1.**
*Let $C_{+}=\left\{\lambda |Re\left(\lambda \right)>0\right\}$ and $C_{+}^{∼}=\left\{\lambda |Re\left(\lambda \right)\geq 0\right\}$. Let F be a complex-valued measurable function on $C_{0}\left[0,T\right]$ such that the integral*
(2)$$J_{F}\left(\lambda \right)=E_{x}(F\left(\lambda^{-\frac{1}{2}}x\right))$$
*exists for all real λ>0. If there exists an analytic function $J_{F}^{*}\left(z\right)$ analytic on $C_{+}$ such that $J_{F}^{*}\left(\lambda \right)=J_{F}\left(\lambda \right)$ for all real λ>0, then we define $J_{F}^{*}\left(z\right)$ to be the analytic Wiener integral of F over $C_{0}\left[0,T\right]$ with parameter z and for each $z\in C_{+}$, we write*
(3)$$E_{x}^{anw_{z}}(F\left(x\right))=E_{x}(F\left(z^{-\frac{1}{2}}x\right))=J_{F}^{*}\left(z\right)$$

*Let q be a non-zero real number and let F be a function whose analytic Wiener integral exists for each z in $C_{+}$. If the following limit exists, then we call it the analytic Feynman integral of F over $C_{0}\left[0,T\right]$ with parameter q, and we write*
(4)$$E_{x}^{anf_{q}}(F\left(x\right))=\lim_{z\to -iq}E_{x}^{anw_{z}}(F\left(x\right)),$$
*where z approaches −iq through $C_{+}$ and $i^{2}=-1$.*


**Definition** **2**(Ref. [16])**.**
*The first variation of a Wiener measurable functional F in the direction $w\in C_{0}\left[0,T\right]$ is defined by the partial derivative:*
(5)$$\delta F\left(x|w\right)=\frac{\partial }{\partial h}{F\left(x+hw\right)|}_{h=0}\;$$

**Remark** **1.**
*We will denote the Formula (5) by $\left(D_{w}F\right)\left(x\right)$ whose notation is motivated from the directional derivative $D_{\vec{u}}f\left(a,b\right)=\lim_{h\to 0}\frac{f\left(a+hu_{1},b+hu_{2}\right)-f\left(a,b\right)}{h}$ in the Calculus and we call $\left(D_{w}F\right)\left(x\right)$ by the directional derivative on the function space $C_{0}\left[0,T\right]$.*


**Theorem** **1**
**(Wiener Integration Formula).**
*Let $F\left(x\right)=f\left(<x,\vec{\alpha }>\right)$, where $f:R^{n}\to C$ is a Lebesgue measurable function on $R^{n}$. Then*
(6)$$\begin{array}{ccc} & & E_{x}(f\left(<x,\vec{\alpha }>\right)\;)={\left(\frac{1}{2\pi }\right)}^{\frac{n}{2}}\int_{R^{n}}f\left(\vec{u}\right)\exp\{-\frac{1}{2}\;||\vec{u}{\left|\right|}^{2}\}d\vec{u} \end{array}$$
*where we set $<x,\vec{\alpha }>=\left(<x,\alpha_{1}>,\cdots ,<x,\alpha_{n}>\right)$ and $<x,\alpha_{j}>=\int_{0}^{T}\alpha_{j}\left(t\right)\;dx\left(t\right)$ is a Paley-Wiener-Zygmund integral for 1≤j≤n and $\left|\right|\vec{u}{\left|\right|}^{2}=\sum_{j=1}^{n}u_{j}^{2}$ and they are equal and $\left\{\alpha_{1},\alpha_{2},\cdots ,\alpha_{n}\right\}$ is an orthonormal class of $L_{2}\left[0,T\right]$.*


**Remark** **2.**
*We will use several times the following formula to prove the main result: For $a\in C_{+}$ and b∈R,*
(7)$$\int_{R}\exp\{-au^{2}+ibu\}\;du=\sqrt{\frac{\pi }{a}}\exp\{-\frac{b^{2}}{4a}\}.$$


## 3. Main Results

Define $F:C_{0}\left[0,T\right]\to C$ by
(8)$$\begin{array}{c} F\left(x\right)=\hat{\mu }(<x,\vec{\alpha }\left(t\right)>), \end{array}$$
where $\left\{\alpha_{1},\alpha_{2},\cdots ,\alpha_{n}\right\}$ is an orthonormal class of $L_{2}\left[0,T\right]$ and
(9)$$\begin{array}{c} \hat{\mu }\left(\vec{u}\right)=\int_{R^{n}}\exp\{i\;(\vec{u}\circ \vec{v}\;)\}\mu \left(d\vec{v}\right),\;\vec{u}\in R^{n} \end{array}$$
is the Fourier transform of the measure μ on $R^{n}$ and $\vec{u}=\left(u_{1},\cdots ,u_{n}\right)$ and $\vec{v}=\left(v_{1},\cdots ,v_{n}\right)$ are in $R^{n}$ and $\vec{u}\circ \vec{v}=\sum_{j=1}^{n}u_{j}v_{j}$.

Because $<x,\vec{\alpha }>=\left(<x,\alpha_{1}>,\cdots ,<x,\alpha_{n}>\right)$ and $<x,\alpha_{j}>=\int_{0}^{T}\alpha_{j}\left(t\right)\;dx\left(t\right)$ for 1≤j≤n, $F\left(x\right)=\hat{\mu }(<x,\vec{\alpha }\left(t\right)>)=\hat{\mu }(\int_{0}^{T}\alpha_{1}\left(t\right)\;dx\left(t\right),\cdots ,\int_{0}^{T}\alpha_{n}\left(t\right)\;dx\left(t\right))$.

Throughout this section, we assume that $w\in C_{0}\left[0,T\right]$ is absolutely continuous in [0,T] with $w^{\prime }\in L_{2}\left[0,T\right]$ and assume that $\int_{R^{n}}(\sum_{j=1}^{n}\;|v_{j}|)|\mu |\left(d\vec{v}\right)<\infty$.

First, we deduce the directional derivative on the function space as a vector calculus form.

**Theorem** **2.**
*The directional derivative on the function space of F(x) exists and*
(10)$$\left(D_{w}F\right)\left(x\right)=\int_{R^{n}}(i<w,\vec{\alpha }>\circ \;\vec{v}\;)\;\exp\{i<x,\vec{\alpha }>\circ \;\vec{v}\;\}\mu \left(d\vec{v}\right)$$


**Proof.** By Definition 2,
(11)$$\begin{array}{ccc} \; & \; & \left(D_{w}F\right)\left(x\right)=\frac{\partial }{\partial h}F\left(x+hw\right){|}_{h=0}\\ \; & = & \frac{\partial }{\partial h}\hat{\mu }(<x+hw,\vec{\alpha }{>)|}_{h=0}\\ \; & = & \frac{\partial }{\partial h}\int_{R^{n}}\exp\{i\;<x+hw\;,\vec{\alpha }>\circ \;\vec{v}\;\}\mu \left(d\vec{v}\right){|}_{h=0}\\ \; & = & \frac{\partial }{\partial h}\int_{R^{n}}\exp\{i\;<x,\vec{\alpha }>\circ \;\vec{v}+i\;h\;<w,\vec{\alpha }>\circ \;\vec{v}\;\}\mu \left(d\vec{v}\right){|}_{h=0}\\ \; & = & \int_{R^{n}}(i<w,\vec{\alpha }>\;\circ \;\vec{v}\;)\;\exp\{\;i\;<x,\vec{\alpha }>\circ \;\vec{v}\;\}\mu \left(d\vec{v}\right)\;. \end{array}$$The Paley-Wiener-Zygmund integral equals to the Riemann Stieltzes integral
$$<w,\alpha_{j}>\;=\int_{0}^{T}\alpha_{j}\left(t\right)\;dw\left(t\right)\;=\;\int_{0}^{T}\alpha_{j}\left(t\right)\;w^{\prime }\left(t\right)\;dt\;,1\leq j\leq n,$$
as *w* is an absolutely continuous function in [0,T] with $w^{\prime }\left(t\right)\in L_{2}\left[0,T\right]$. Therefore,
(12)$$\begin{array}{ccc} \; & \; & \left(D_{w}F\right)\left(x\right)\\ \; & = & \int_{R^{n}}\;(\;i<w,\vec{\alpha }>\;\circ \;\vec{v}\;)\;\exp\{\;i\;<x,\vec{\alpha }>\;\circ \;\vec{v}\;\}\mu \left(d\vec{v}\right)\\ \; & = & \int_{R^{n}}(i\;\sum_{j=1}^{n}(\int_{0}^{T}\alpha_{j}\left(t\right)\;dw\left(t\right))\;v_{j}\left(t\right))\;\exp\{i\sum_{j=1}^{n}(\int_{0}^{T}\alpha_{j}\left(t\right)\;dx\left(t\right))\;v_{j}\left(t\right)\;\}\mu \left(d\vec{v}\right)\\ \; & = & \int_{R^{n}}(i\;\sum_{j=1}^{n}(\int_{0}^{T}\alpha_{j}\left(t\right)\;w^{\prime }\left(t\right)dt)\;v_{j}\left(t\right))\exp\{i\sum_{j=1}^{n}(\int_{0}^{T}\alpha_{j}\left(t\right)\;dx\left(t\right))\;v_{j}\left(t\right)\}\mu \left(d\vec{v}\right)\;\; \end{array}$$
and
(13)$$\begin{array}{ccc} \left|\left(D_{w}F\right)\left(x\right)\right| & \leq & \int_{R^{n}}\;|\sum_{j=1}^{n}(\int_{0}^{T}\alpha_{j}\left(t\right)\;w^{\prime }\left(t\right)dt)\;v_{j}\left(t\right)\;|\;\left|\mu \right|\left(d\vec{v}\right)\\ \; & \leq & \int_{R^{n}}\;\sum_{j=1}^{n}\;(\;(||\;\alpha_{j}{\;||}_{2}\;\times \;||w^{\prime }{\left|\right|}_{2})\times \;|v_{j}\left|\;)\;|\mu \right|\left(d\vec{v}\right)\\ \; & = & \left|\right|w^{\prime }{\left|\right|}_{2}\;\int_{R^{n}}\;(\sum_{j=1}^{n}\;|v_{j}\left|\;)|\mu \right|\left(d\vec{v}\right)\\ \; & < & \infty , \end{array}$$
by a Hölder inequality in $\;L_{2}\left[0,T\right]$. Therefore $\left(D_{w}F\right)\left(x\right)$ exists. □

In the next Theorem, we obtain the analytic Wiener integral of $\left(D_{w}F\right)\left(x\right)$ on the function space as a vector calculus form:

**Theorem** **3.**
*For every $z\in C_{+}$,*
(14)$$\begin{array}{ccc} & & E_{x}^{anw_{z}}(\left(D_{w}F\right)\left(x\right))=\int_{R^{n}}(i\;<w,\vec{\alpha }>\;\circ \;\vec{v}\;)\exp\{-\frac{1}{2z}\left|\right|\vec{v}{\left|\right|}^{2}\;\}\mu \left(d\vec{v}\right)\;\; \end{array}$$


**Proof.** By (12), we have that for $z\in C_{+}$,
(15)$$\begin{array}{ccc} \; & \; & E_{x}^{anw_{z}}(\left(D_{w}F\right)\left(x\right))\\ \; & = & E_{x}(\left(D_{w}F\right)\left(z^{-\frac{1}{2}}x\right))\\ \; & = & E_{x}(\;\int_{R^{n}}(i\;<w,\vec{\alpha }>\;\circ \;\vec{v})\exp\{i\;z^{-\frac{1}{2}}<x,\vec{\alpha }>\;\circ \;\vec{v}\;\}\mu \left(d\vec{v}\right))\\ \; & = & E_{x}(\int_{R^{n}}(i<w,\vec{\alpha }>\;\circ \;\vec{v}\;)\exp\{i\;z^{-\frac{1}{2}}\sum_{j=1}^{n}\;(\int_{0}^{T}\alpha_{j}\left(t\right)\;dx\left(t\right))\;v_{j}\left(t\right)\}\mu \left(d\vec{v}\right))\\ \; & = & {\left(\frac{1}{2\pi }\right)}^{\frac{n}{2}}\int_{R^{n}}\;[\int_{R^{n}}(i\;<w,\vec{\alpha }>\;\circ \;\vec{v}\;)\exp\{i\;z^{-\frac{1}{2}}\;\sum_{j=1}^{n}\;(u_{j}\;\cdot v_{j})\}\mu \left(d\vec{v}\right)]\exp\{-\frac{1}{2}\sum_{j=1}^{n}u_{j}^{2}\}d\vec{u}\\ \; & = & {\left(\frac{1}{2\pi }\right)}^{\frac{n}{2}}\int_{R^{n}}(i\;<w,\vec{\alpha }>\;\circ \;\vec{v})[\int_{R^{n}}\exp\{\sum_{j=1}^{n}\;\left(\;-\frac{1}{2}{u_{j}}^{2}+i\;z^{-\frac{1}{2}}u_{j}\;v_{j}\right)\}d\vec{u}\;]\mu \left(d\vec{v}\right)\\ \; & = & {\left(\frac{1}{2\pi }\right)}^{\frac{n}{2}}\int_{R^{n}}(i\;<w,\vec{\alpha }>\;\circ \;\vec{v})\;[\;{\left(2\pi \right)}^{\frac{n}{2}}\exp\{-\frac{1}{2z}\sum_{j=1}^{n}v_{j}^{2}\;\}]\mu \left(d\vec{v}\right)\\ \; & = & \int_{R^{n}}(i\;<w,\vec{\alpha }>\;\circ \;\vec{v})\exp\{-\frac{1}{2z}\left|\right|\vec{v}{\left|\right|}^{2}\}\mu \left(d\vec{v}\right)\;. \end{array}$$By (13), we have
(16)$$\begin{array}{ccc} \; & \; & \left|E_{x}^{anw_{z}}(\left(D_{w}F\right)\left(x\right))\right|\\ \; & \leq & \left|\int_{R^{n}}(i\;<w,\vec{\alpha }>\;\circ \;\vec{v})\exp\{-\frac{1}{2z}\left|\right|\vec{v}{\left|\right|}^{2}\}\mu \left(d\vec{v}\right)\right|\\ \; & \leq & \left|\int_{R^{n}}(i\;<w,\vec{\alpha }>\;\circ \;\vec{v})\mu \left(d\vec{v}\right)\right|\\ \; & \leq & \int_{R^{n}}\;|\sum_{j=1}^{n}(\int_{0}^{T}\alpha_{j}\left(t\right)\;w^{\prime }\left(t\right)dt)\;v_{j}\left(t\right)\;|\;\left|\mu \right|\left(d\vec{v}\right)\\ \; & \leq & \int_{R^{n}}\;\sum_{j=1}^{n}\;(\;(||\;\alpha_{j}{\;||}_{2}\;\times \;||w^{\prime }{\left|\right|}_{2})\times \;|v_{j}\left|\;)\;|\mu \right|\left(d\vec{v}\right)\\ \; & = & \left|\right|w^{\prime }{\left|\right|}_{2}\;\int_{R^{n}}\;(\sum_{j=1}^{n}\;|v_{j}|\;)|\mu |\left(d\vec{v}\right)\\ \; & < & \infty . \end{array}$$ □

To prove the relationship between the function space integral and the directional derivative on the functions space, we have to prove the following theorem:

**Theorem** **4.**
*For $z\in C_{+}$,*
(17)$$\begin{array}{ccc} & & \exp\{\frac{1-z}{2}\;\;||<x,\vec{\alpha }>{\left|\right|}^{2}\;\}\;\left(D_{w}F\right)\left(x\right) \end{array}$$
*is a Wiener integrable function of $x\in C_{0}\left[0,T\right]$.*


**Proof.** By Equation (Equation 6),
(18)$$\begin{array}{ccc} \; & \; & E_{x}(\exp\{\frac{1-z}{2}\;||<x,\vec{\alpha }>{\left|\right|}^{2}\;\}\;\left(D_{w}F\right)\left(x\right))\\ \; & = & E_{x}(\exp\{\frac{1-z}{2}\sum_{j=1}^{n}(\int_{0}^{T}\alpha_{j}\left(t\right)\;dx\left(t\right){)}^{2}\}\int_{R^{n}}(i<w,\vec{\alpha }>\circ \;\vec{v})\\ \; & \; & \;\times \;\exp\{i<x,\vec{\alpha }>\circ \;\vec{v}\;\}\mu \left(d\vec{v}\right))\\ \; & = & \int_{R^{n}}(i\;<w,\vec{\alpha }>\;\circ \;\vec{v}\;)\\ \; & \; & \;\times E_{x}(\exp\{\sum_{j=1}^{n}\frac{1-z}{2}(\int_{0}^{T}\alpha_{j}\left(t\right)\;dx\left(t\right){)}^{2}\;+\;i\;\sum_{j=1}^{n}(\int_{0}^{T}\alpha_{j}\left(t\right)\;dx\left(t\right))\;v_{j}\left(t\right)\})\mu \left(d\vec{v}\right)\\ \; & = & \int_{R^{n}}(i\;<w,\vec{\alpha }>\circ \;\vec{v}\;)\\ \; & \; & \;\times [{\left(\frac{1}{2\pi }\right)}^{\frac{n}{2}}\int_{R^{n}}\exp\{\sum_{j=1}^{n}\frac{1-z}{2}{u_{j}}^{2}+\;i\;u_{j}v_{j}\}\;\exp\{-\frac{1}{2}\sum_{j=1}^{n}{u_{j}}^{2}\}d\vec{u}]\mu \left(d\vec{v}\right)\\ \; & = & {\left(\frac{1}{2\pi }\right)}^{\frac{n}{2}}\int_{R^{n}}(i\;<w,\vec{\alpha }>\circ \vec{v}\;)[\int_{R^{n}}\exp\{\sum_{j=1}^{n}\left(-\frac{z}{2}{u_{j}}^{2}+i\;v_{j}\;u_{j}\right)\}d\vec{u}]\mu \left(d\vec{v}\right)\\ \; & = & {\left(\frac{1}{2\pi }\right)}^{\frac{n}{2}}\int_{R^{n}}(i\;<w,\vec{\alpha }>\circ \;\vec{v}\;)[(\frac{2\pi }{z}{)}^{\frac{n}{2}}\exp\{-\frac{1}{2z}\sum_{j=1}^{n}{v_{j}}^{2}\}]\mu \left(d\vec{v}\right)\\ \; & = & z^{-\frac{n}{2}}\;\int_{R^{n}}(i\;<w,\vec{\alpha }>\;\circ \;\vec{v})\exp\{-\frac{1}{2z}\;||\vec{v}{\left|\right|}^{2}\}\mu \left(d\vec{v}\right)\;, \end{array}$$
and
(19)$$\begin{array}{ccc} \; & \; & \left|\;z^{-\frac{n}{2}}\;\int_{R^{n}}(i\;<w,\vec{\alpha }>\;\circ \;\vec{v})\exp\{-\frac{1}{2z}\;||\vec{v}{\left|\right|}^{2}\}\;\mu \left(d\vec{v}\right)\;\right|\\ \; & \leq & z^{-\frac{n}{2}}\;\int_{R^{n}}\;|\;\sum_{j=1}^{n}(\int_{0}^{T}\alpha_{j}\left(t\right)\;dw\left(t\right))\;v_{j}\left(t\right)\;|\;\left|\mu \right|\left(d\vec{v}\right)\\ \; & = & z^{-\frac{n}{2}}\;\int_{R^{n}}\;|\;\sum_{j=1}^{n}(\int_{0}^{T}\alpha_{j}\left(t\right)\;w^{\prime }\left(t\right)\;dt)\;v_{j}\left(t\right)\;|\;\left|\mu \right|\left(d\vec{v}\right)\\ \; & \leq & z^{-\frac{n}{2}}\;\sum_{j=1}^{n}\left[\;(|\right|\alpha_{j}{\left|\right|}_{2}\times \left|\right|w^{\prime }{\left|\right|}_{2})\times |v_{j}\left|]\;|\mu \right|\left(d\vec{v}\right)\\ \; & = & z^{-\frac{n}{2}}\;||w^{\prime }{\left|\right|}_{2}\;\int_{R^{n}}(\sum_{j=1}^{n}\;|v_{j}|)\;|\mu |\left(d\vec{v}\right)\\ \; & < & \infty . \end{array}$$Therefore, the function in (17) is a Wiener integrable function of $x\in C_{0}\left[0,T\right]$. □

Now, we prove that the analytic Wiener integral of the directional derivative on the function space is expressed as the sequence of Wiener integrals and we express the formula as a vector calculus form:

**Theorem** **5.**
*For $z\in C_{+}$,*
(20)$$\begin{array}{ccc} & & E_{x}^{anw_{z}}(\left(D_{w}F\right)\left(x\right))=z^{\frac{n}{2}}\;E_{x}(\exp\{\frac{1-z}{2}\;||<x,\vec{\alpha }>{\left|\right|}^{2}\}\;\left(D_{w}F\right)\left(x\right)). \end{array}$$


**Proof.** By Theorems 3 and 4,
(21)$$\begin{array}{ccc} \; & \; & E_{x}(\exp\{\frac{1-z}{2}\;||<x,\vec{\alpha }>{\left|\right|}^{2}\}\;\left(D_{w}F\right)\left(x\right))\\ \; & = & z^{-\frac{n}{2}}\;\int_{R^{n}}\;(i<w,\vec{\alpha }>\circ \;\vec{v}\;)\exp\{-\frac{1}{2z}\;||\vec{v}{\left|\right|}^{2}\}\mu \left(d\vec{v}\right)\\ \; & = & z^{-\frac{n}{2}}\;E_{x}^{anw_{z}}(\left(D_{w}F\right)\left(x\right)). \end{array}$$ □

Now, we prove that the directional derivative on the function space satisfies the change of scale formula for the function space integral and we express the formula as a vector calculus form:

**Theorem** **6**
**(Change of scale formula).**
*For real ρ>0,*
(22)$$\begin{array}{ccc} & & E_{x}(\left(D_{w}F\right)\left(x\right))=\rho^{-n}\;E_{x}(\exp\{\frac{\rho^{2}-1}{2\rho^{2}}\;\;||<x,\vec{\alpha }>{\left|\right|}^{2}\}\;\left(D_{w}F\right)\left(x\right))\; \end{array}$$


**Proof.** By Theorem 5, we have that for real z>0,
(23)$$\begin{array}{ccc} \; & \; & E_{x}^{anw_{z}}(\left(D_{w}F\right)\left(x\right))\\ \; & = & E_{x}(\left(D_{w}F\right)\left(z^{-\frac{1}{2}}x|w\right))\\ \; & = & z^{\frac{n}{2}}\;E_{x}(\exp\{\frac{1-z}{2}\;\;||<x,\vec{\alpha }>{\left|\right|}^{2}\}\;\left(D_{w}F\right)\left(x\right)) \end{array}$$Taking $z=\rho^{-2}$, we have (23). □

Now, we prove that the analytic Feynman integral of the directional derivative on the function space exists and we express it as a vector calculus form:

**Theorem** **7.**
(24)$$\begin{array}{ccc} & & E_{x}^{anf_{q}}(\left(D_{w}F\right)\left(x\right))=\int_{R^{n}}\;(i<w,\vec{\alpha }>\;\circ \;\vec{v}\;)\exp\{-\frac{i}{2q}\left|\right|\vec{v}{\left|\right|}^{2}\}\mu \left(d\vec{v}\right) \end{array}$$


**Proof.** By Theorem 3,
(25)$$\begin{array}{ccc} \; & \; & E_{x}^{anf_{q}}(\left(D_{w}F\right)\left(x\right))\\ \; & = & \lim_{z\to -iq}\;E_{x}^{anw_{z}}(\left(D_{w}F\right)\left(x\right))\\ \; & = & \lim_{z\to -iq}\int_{R^{n}}(i<w,\vec{\alpha }>\;\circ \;\vec{v}\;)\exp\{-\frac{1}{2z}\;||\vec{v}{\left|\right|}^{2}\}\mu \left(d\vec{v}\right)\\ \; & = & \int_{R^{n}}\;(<w,\vec{\alpha }>\circ \;\vec{v}\;)\exp\{-\frac{i}{2q}\;||\vec{v}{\left|\right|}^{2}\}\mu \left(d\vec{v}\right) \end{array}$$
whenever z→−iq through $C_{+}$. By (16) and by (25), we have
(26)$$\begin{array}{ccc} \; & \; & \left|\;E_{x}^{anf_{q}}(\left(D_{w}F\right)\left(x\right))\;\right|\\ \; & \leq & \int_{R^{n}}|<w,\vec{\alpha }\left(t\right)>\circ \;\vec{v}\;|\;\left|\mu \right|\left(d\vec{v}\right)\\ \; & = & \int_{R^{n}}|\sum_{j=1}^{n}[(\int_{0}^{T}\alpha_{j}\left(t\right)\;dw\left(t\right))\times \;|v_{j}\left(t\right)|]\;|\;|\mu |\left(d\vec{v}\right)\\ \; & \leq & \left|\right|w^{\prime }{\left|\right|}_{2}\;\int_{R^{n}}(\sum_{j=1}^{n}\;|v_{j}\left|)\;|\mu \right|\left(d\vec{v}\right)\\ \; & < & \infty \;. \end{array}$$ □

Finally, we prove that the analytic Feynman integral of the directional derivative on the function space is expressed as the sequence of Wiener integrals of the directional derivative on the function space and we express the formula as a vector calculus form:

**Theorem** **8.**
(27)$$\begin{array}{ccc} & & E_{x}^{anf_{q}}(\left(D_{w}F\right)\left(x\right))=\lim_{k\to \infty }\;{z_{k}}^{\frac{n}{2}}\;E_{x}(\exp\{\frac{1-z_{k}}{2}\;\;||<x,\vec{\alpha }>{\left|\right|}^{2}\}\;\left(D_{w}F\right)\left(x\right)) \end{array}$$
*whenever $\{z_{k}\}\;\to -\;iq$ through $C_{+}$.*


**Proof.** By Theorem 5,
(28)$$\begin{array}{ccc} \; & \; & E_{x}^{anf_{q}}(\left(D_{w}F\right)\left(x\right))\\ \; & = & \lim_{k\to \infty }E_{x}^{anw_{z_{k}}}(\left(D_{w}F\right)\left(x\right))\\ \; & = & \lim_{k\to \infty }\;\;\;{z_{k}}^{\frac{n}{2}}\;E_{x}(\exp\{\frac{1-z_{k}}{2}\;\;||<x,\vec{\alpha }>{\left|\right|}^{2}\}\;\left(D_{w}F\right)\left(x\right)) \end{array}$$
whenever $\{z_{k}\}\;\to -\;iq$ through $C_{+}$. □

## 4. Conclusions

In this paper, we find a new expression of the vector calculus approach to the change of scale formula for the Wiener integral (which is motivated from the Heat Equaton in Quantum Mechanics) about the directional derivative on the function space of a Fourier transform.

**Remark** **3.**
*Notations and Theorems of this paper are upgraded from the reviewer’s comment. The author is very grateful to reviewers.*

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
