# Peer review of "A Partial Derivative Approach to the Change of Scale Formula for the Function Space Integral"

_entropy, 2020, doi:10.3390/e23010026_

Round 1

Reviewer 1 Report

This paper is very technical, written only for specialists.  The introduction is sparce in content, indicating very little.  Also, basic references to basic results are missing. These should be included. There is little discussion of theorems and their meaning, as they are listed and proved one after the other, theorem-proof. The proofs are long calculations with little explanation of why anyone would want such extensions.

Please not I am not an expert on this topic. Therefore it seems excessively technical with little explanation of why the results are significant. 

Author Response

Please see file attached.

Reviewer 2 Report

The author extends some existing results on the Wiener and Feynman integrals related to scale change.  The presentation is rigorous.  It consists of a sequence of theorems and proofs.  The author provides no motivation for the study.  It is not clear whether the results in the paper are useful in any application.  For this reason, this reviewer believes that Entropy is not a good home for this paper.

Author Response

Please see file attached.

Reviewer 3 Report

The author derives several formulas involving "first variations" (or directed derivatives) of Wiener-measurable functions.

The formulas seem correct, and thus the paper seems suitable for publication (although the content of the article seem to have no relation to entropy; but i leave to the editor to determine how this aspect matters).

The author could optionally
. comment on the series of theorems 3.1-3.7, and
. put more context on the motivations for deriving them (in the introduction), and for their interest (in the conclusion for instance).

Minor remark: some sentences are difficult to understand (in §3 in particular), and typos can be found in the text ; the paper should be proof-read by a native English speaker.

Author Response

Please see file attached.
